# Anthocyanin Accumulation Provides Protection against High Light Stress While Reducing Photosynthesis in Apple Leaves

**DOI:** 10.3390/ijms232012616

**Published:** 2022-10-20

**Authors:** Shanshan Zhao, Jeremie A. Blum, Fangfang Ma, Yuzhu Wang, Ewa Borejsza-Wysocka, Fengwang Ma, Lailiang Cheng, Pengmin Li

**Affiliations:** 1State Key Laboratory of Crop Stress Biology for Arid Areas/Shaanxi Key Laboratory of Apple, College of Horticulture, Northwest A&F University, Xianyang 712100, China; 2Horticulture Section, School of Integrative Plant Science, Cornell University, Ithaca, NY 14853, USA; 3Donald Danforth Plant Science Center and Agricultural Research Service, US Department of Agriculture, St. Louis, MO 63132, USA; 4State Key Laboratory of Crop Biology, College of Horticulture Science and Engineering, Shandong Agricultural University, Tai’an 271018, China

**Keywords:** anthocyanin, apple, high light, light attenuation, photoprotection, photosynthesis

## Abstract

The photoprotective role of anthocyanin remains controversial. In this study, we explored the effects of anthocyanin on photosynthesis and photoprotection using transgenic ‘Galaxy Gala’ apple plants overexpressing *MdMYB10* under high light stress. The overexpression of *MdMYB10* dramatically enhanced leaf anthocyanin accumulation, allowing more visible light to be absorbed, particularly in the green region. However, through post-transcriptional regulation, anthocyanin accumulation lowered leaf photosynthesis in both photochemical reaction and CO_2_ fixation capacities. Anthocyanin accumulation also led to a decreased de-epoxidation state of the xanthophyll cycle and antioxidant capacities, but this is most likely a response to the light-shielding effect of anthocyanin, as indicated by a higher chlorophyll concentration and lower chlorophyll a/b ratio. Under laboratory conditions when detached leaves lost carbon fixation capacity due to the limitation of CO_2_ supply, the photoinhibition of detached transgenic red leaves was less severe under strong white, green, or blue light, but it became more severe in response to strong red light compared with that of the wild type. In field conditions when photosynthesis was performed normally in both green and transgenic red leaves, the degree of photoinhibition was comparable between transgenic red leaves and wild type leaves, but it was less severe in transgenic young shoot bark compared with the wild type. Taken together, these data show that anthocyanin protects plants from high light stress by absorbing excessive visible light despite reducing photosynthesis.

## 1. Introduction

Anthocyanins comprise a group of water-soluble pigments synthesized by the flavonoid metabolic pathway and are responsible for the red, purple, and blue colors of plant leaves, fruits, flowers, and other tissues. The biosynthesis of anthocyanin begins with 4-coumaroyl-CoA and is catalyzed by key enzymes, including chalcone synthase (CHS), chalcone isomerase (CHI), flavanone 3-hydroxylase (F3H), flavonoid 3′-hydroxylase (F3′H), dihydroflavonol 4-reductase (DFR), anthocyanin synthase (ANS), and UDP-glycose: flavonoid glycosyltransferase (UFGT). In most plants, the MYB-bHLH-WD40 complex regulates the expression of genes encoding these enzymes [1,2,3,4]. After synthesis, anthocyanin is transported and stored in vacuoles.

Anthocyanin has an absorption spectrum in visible light, especially in the green light region. Like red and blue light, green light may also efficiently drive photosynthesis, especially at a high photosynthetic photon flux density [5,6]. Therefore, it is generally accepted that epidermal anthocyanin may protect plants from photoinhibition/photodamage by lowering the light intensity irradiation of chloroplasts under high light stress [7,8,9,10]. However, in approximately 30% of published reports, the physiological responses of red and green leaves to high light were either comparable, or red leaves performed worse than their green counterparts [7]. The difference in genetic backgrounds between green and red plants might partly account for this inconsistency [7,11]. However, it should be noted that green light can regulate the expression of many genes, as well as the growth and development of plants [9,12,13,14,15]. Attenuation of blue light by anthocyanin may also alter gene expression in plants [11]. Thus, in addition to reducing the light intensity irradiation of plant cells, anthocyanin accumulation might also affect plant resistance to high light stress by changing the light quality.

Under high light stress, the absorbed light energy cannot be consumed in a timely manner by photosynthesis, and photoprotective mechanisms, such as thermal dissipation, are enhanced to consume the excess excited light energy [16,17]. If the excess excited light energy is still not dissipated in a timely manner, then the production of reactive oxygen species (ROS) may be triggered, which are then scavenged by antioxidant systems to reduce the photooxidative stress [18,19]. Therefore, if anthocyanin has certain negative effects on photosynthesis, thermal dissipation, antioxidant systems, or other photoprotective mechanisms, it may counteract the positive effect via anthocyanin shielding and the subsequent resistance of plants to high light stress. Indeed, relatively lower levels of key photosynthetic enzymes and photoprotective capacities have been reported in red plants compared with their green counterparts [20,21,22,23]. However, the effects of anthocyanin on photosynthesis and photoprotective mechanisms remain controversial, because photosynthesis, the xanthophyll cycle, or antioxidant systems were also found to be comparable or even enhanced in red tissues [7,11,24,25].

As anthocyanin may react with oxidants or ROS in vitro, it is implied that anthocyanin may also scavenge ROS in vivo to protect plants from oxidative stress [23,26,27]. Nevertheless, unlike anthocyanin, which is only present in certain specialized plants, antioxidant systems, such as the ascorbate-glutathione cycle, are ubiquitous in the plant kingdom. Moreover, compared with anthocyanin, which is mainly localized in vacuoles, the antioxidant system is situated in more direct positions against the excess accumulation of ROS [18,19,20,25]. It has been reported that chloroplast-derived ROS are not associated with anthocyanin accumulation under high light stress [28]. Whether anthocyanins may scavenge ROS in vivo is still debated. 

In recent years, red-flesh apple breeding programs have attracted more attention. Usually, the leaves of red-flesh apple genotypes are also rich in anthocyanins. However, it is unclear whether the accumulated anthocyanin may affect photosynthesis or photoinhibition of apple leaves. In the present study, the ‘Galaxy Gala’ apple plant and its transgenic lines with abundant anthocyanin, which share the same genetic background with the exception of anthocyanin, were used to explore the effects of anthocyanin on photosynthesis and photoprotection under high light stress.

## 2. Results

### 2.1. Effects of MdMYB10 Overexpression on Anthocyanin Accumulation

The overexpression of MdMBY10 enhanced the expression of chalcone synthase (MdCHS), chalcone isomerase (MdCHI), flavanone 3-hydroxylase (MdF3H), flavonoid 3′-hydroxylase (MdF3′H), dihydroflavonol 4-reductase (MdDFR), anthocyanin synthase (MdANS), UDP-glycose: flavonoid glycosyltransferase (MdUFGT), and glutathione S-transferase (MdGST), the key genes involved in anthocyanin synthesis and transfer in transgenic leaves (Figure 1). As a result, the overexpression of MdMYB10 caused the leaves to accumulate abundant anthocyanin content and become red. Regarding the other flavonoid compounds, the content of phlorizin was not altered, whereas that of flavonol was increased by approximately 30% in MdMYB10 overexpression leaves (Appendix A). Moreover, quercetin-3-O-glycosides showed distinct change patterns in transgenic leaves, with the contents of three of them (quercetin-3-O-galactoside, quercetin-3-O-xyloside, and quercetin-3-O-arabinoside) increasing while two of them (quercetin-3-O-glucoside and quercetin-3-O-rhamnoside) decreased. The overexpression of MdMYB10 also increased the accumulation of anthocyanin in young branch barks (Appendix A).

### 2.2. Effects of Anthocyanin Accumulation on Photosynthesis

Dark respiratory rate was comparable between WT and transgenic red leaves (Figure 2A). When the photon flux density (PFD) was higher than 150 μmol m^−2^ s^−1^, the net photosynthetic rate was significantly lower in transgenic red leaves compared with WT leaves (Figure 2A). Under saturated PFD, the photosynthetic rate was approximately 70% in transgenic compared with WT plants. Apparent quantum yield was also lower in transgenic red leaves. With the increase in intercellular CO_2_ concentration, photosynthetic rates increased in transgenic and WT leaves (Figure 2B). When the intercellular CO_2_ concentration was approximately 500 μmol mol^−1^, the photosynthetic rates reached the maximum in transgenic and WT leaves. Apparent carboxylation efficiency was significantly lower in transgenic compared with WT plants.

Based on the leaf reflection spectra, *MdMYB10* overexpression leaves showed more absorption in visible light, especially in green light (550–600 nm, Figure 3A). Stomatal phenotypes were similar between transgenic and WT plants (Appendix A). The synthesis of anthocyanin increased the contents of chlorophylls, especially chlorophyll b. As a result, the chlorophyll a/b ratio decreased in transgenic red leaves (Figure 3B).

The chlorophyll a fluorescence transient and the 820 nm modulated reflection curves were also altered by the accumulation of anthocyanin (Figure 4). Compared with WT, transgenic leaves showed higher fluorescence signals from the O to I step but a slower decrease in the 820 nm reflection from MRo to MRmin, indicating that the photosynthetic electron transport capacity between the two photosystems was lower in transgenic red leaves [29].

Regarding the differentially expressed genes between transgenic and WT leaves, KEGG enrichment analysis revealed that many pathways were affected by anthocyanin synthesis, including primary and secondary metabolism, transcription, translation, membrane transport, and signal transduction, among others (Appendix A). However, none of the gene involved in photosynthesis showed different expression levels between transgenic and WT leaves (Appendix A). Interestingly, the activities of Rubisco, GAPDH, PRK, A6PR and SPS, key enzymes involved in photosynthesis, were significantly lower in *MdMYB10* overexpression leaves (Figure 5; Appendix A). To further assay the effects of anthocyanin accumulation on photosynthesis, a targeted metabolomics assay was performed via LC-MS/MS to quantify the changes in metabolites directly involved in the Calvin cycle. The produced photosynthates, including sorbitol, sucrose, and intermediate products such as 3-phosphoglycerate, sorbitol-6-phosphate, sucrose-6-phosphate, adenosine diphosphate glucose, ribulose-1,5-bisphosphate, and others, were all decreased in transgenic red leaves compared with WT leaves (Figure 5; Appendix A). Thus, the decreased contents of these metabolites further confirmed that the overall photosynthetic capacity was impaired in the transgenic lines.

Concerning the expression of genes involved in respiration and other carbohydrate metabolism, 23 genes were upregulated, and 11 were downregulated (Figure 5, Appendix A). With respect to the levels of genes involved in amino acid metabolism, 15 genes were upregulated, and 8 were downregulated. For metabolites involved in carbon metabolism, the contents of 15 kinds of non-structural carbohydrates and organic acids were decreased, including 2-phosphoglycerate, aconitate, 2-oxoglutarate, malate, myo-inositol, glucose, ribose, erythritol, phosphoenolpyruvate, glycerate, xylitol, isocitrate, shikimate, citrate, and succinate, while those of the other 12 kinds of metabolites, such as xylulose, pyruvate, maltose, galactose, maltitol, fumarate, rhamnose, quinate, maleic acid, fructose, or xylose, remained unchanged in transgenic red leaves compared with WT leaves (Figure 5, Appendix A). For nitrogen metabolism, nine kinds of free amino acids had reduced contents, including histidine, glycine, proline, isoleucine, γ-aminobutyrate, leucine, phenylalanine, serine, and alanine, whereas the contents of glutamine and asparagine were significantly higher in transgenic red leaves. Other amino acids, such as methionine, valine, arginine, tyrosine, threonine, glutamate, ornithine, and aspartate, showed comparable contents in WT and transgenic leaves.

### 2.3. Effects of Anthocyanin Accumulation on Photoprotection

Without high light treatment, the accumulation of anthocyanin did not affect the xanthophyll cycle pool size and its de-epoxidation (Figure 6). Upon exposure to strong red or green light, the pool size remained unchanged, whereas the de-epoxidation level was significantly lower in transgenic red leaves compared with WT leaves.

The accumulation of anthocyanin did not affect the activities of APX, MDHAR, and DHAR (Figure 7). However, the activities of SOD and GR and the contents of ascorbate and glutathione were decreased in transgenic compared with WT leaves. Under strong red or green light treatment, the activities of antioxidant enzymes and the contents of ascorbate and glutathione changed in similar patterns in WT and transgenic red leaves, except that the activity of MDHAR showed a greater decrease in transgenic leaves following exposure to red light.

The stomata of apple leaves are mainly distributed on the abaxial side. In this study, when treated under different high light stresses, the abaxial surface of apple leaves was closely attached to wet gauze. In this way, the stomata were sealed with water and the carbon fixation capacity of detached leaves was almost fully inhibited due to the limitation of CO_2_ supply (Under this condition, the Fv/Fm values decreased in similar patterns in transgenic red leaves and WT leaves with strong UV-B and UV-A irradiation (Figure 8A). However, the values of Fv/Fm decreased at a slower rate in transgenic red leaves compared with WT leaves upon exposure to strong white, green, or blue light (Figure 8A and Appendix A). Moreover, the values of Fv/Fm decreased at a faster rate in red leaves following exposure to strong red light. 

In field conditions, the values of Fv/Fm were slightly but significantly higher in red leaves than in WT leaves in the morning (Figure 8B). However, the values became comparable between WT and transgenic leaves in the afternoon. For young branch barks, the values of Fv/Fm were higher in transgenic red barks than in WT in the morning (Figure 8C). In the afternoon, the values of Fv/Fm remained unchanged in transgenic red barks but decreased significantly in WT.

## 3. Discussion

The findings of the present study clearly showed that anthocyanin played two opposing roles in photoprotection under high light stress. On the one hand, anthocyanin accumulation may shade chloroplasts by absorbing visible light, especially green light (Figure 3A), causing leaves to absorb less light energy and thus having a positive effect on photoprotection. On the other hand, the accumulation of anthocyanin could also decrease photosynthesis and the thermal dissipation dependent on the xanthophyll cycle, especially the former (Figure 2, Figure 4, Figure 5 and Figure 6), the major pathways in the consumption of absorbed light energy. This phenomenon had a negative effect on photoprotection. In field conditions when photosynthesis performed normally in both green and transgenic red leaves, the positive and negative effects of anthocyanin on photoprotection might be comparable. The photoinhibition degree was then similar in green and red leaves (Figure 8B). Under laboratory conditions, as carbon fixation capacity was almost fully inhibited in detached leaves due to the limitation of CO_2_ supply, the positive effect of anthocyanin was far greater than the negative effect. Consequently, light attenuation by anthocyanin could alleviate the degree of photoinhibition under white, green, or blue light stress conditions (Figure 8A). Under red light stress without any light attenuation by anthocyanin, the lower thermal dissipation in red leaves (Figure 6) might partly account for the more severe photoinhibition in transgenic red leaves in contrast to WT (Figure 8A). The decreased thermal dissipation was most likely a response to the light-shielding effect of anthocyanin, as indicated by higher chlorophyll concentration and lower chlorophyll a/b ratio (Figure 3). Young branch bark contains chlorophylls and has a normal photochemical reaction capacity. However, the carbon fixation capacity of branch bark is very low similar to other non-leaf green tissues. Under field conditions, the observation that exposed young branch barks of *MdMYB10* overexpression plants lacked any photoinhibition in contrast to WT (Figure 8C) also supports the two opposing roles of anthocyanin in photoprotection under high light stress.

Compared with that in their green counterpart, the reduced photosynthesis in transgenic red leaves might be related to the lower photochemical reaction and CO_2_ fixation capacities (Figure 2, Figure 4 and Figure 5). A stomatal effect could be excluded, as the phenotypes of stomata were similar between the transgenic and WT leaves (Appendix A). In previous studies, it has been reported that green light might regulate gene expression in plants [12,13]. Moreover, anthocyanin content can also alter the gene expression in plants by attenuating blue light [11]. While purple leaves display an overexpression of genes that promote chlorophyll biosynthesis and light harvesting, genes involved in the stability/repair of photosystems were substantial overexpressed in green leaves [11]. Compared with that in WT, the changes in the expression of more than 1000 genes in transgenic red leaves (Appendix A) could be mainly attributed to the visible light attenuation by anthocyanin. Moreover, the comparable expression of genes involved in photosynthesis between transgenic and WT leaves (Appendix A) indicated that anthocyanin accumulation might decrease photosynthesis in apple leaves by regulating photosynthetic genes at the post-transcriptional level. Indeed, the KEGG enrichment analysis revealed that expressions of transcription, translation, membrane transport, and signal transduction genes were changed in the transgenic red leaves (Appendix A). The lower production of photosynthates resulted in lower contents of carbon and nitrogen metabolites in transgenic red leaves (Figure 5), as both carbon and nitrogen metabolism depend on the carbon skeleton supply from photosynthates. The greater accumulation of glutamine and asparagine in transgenic red leaves (Figure 5) further support this notion, as glutamine and asparagine have a relatively lower carbon/nitrogen ratio compared with glutamate and aspartate. Previous studies have also found that anthocyanin accumulation increases the asparagine content in pear peel [30].

Using Arabidopsis and mutants deficient in anthocyanin synthesis, it has been reported that the light attenuation role of anthocyanin is more important than its antioxidant role in photoprotection under long-term high light treatment [10]. In the present study, although anthocyanin accumulation decreased the antioxidant capacity in apple leaves (Figure 7), it might not be involved in ROS scavenging in vivo under high light stress, as the antioxidant systems changed in similar patterns in transgenic red leaves and WT leaves after green or red high light treatment (Figure 7). This is a reasonable supposition as the major production of ROS occurs at the chloroplast under high light stress [19], while anthocyanin is mainly stored in vacuoles.

In previous studies, flavonoids were shown to protect plants from strong UV irradiation stress [31,32]. In the present study, however, the accumulated anthocyanin neither changed the leaf reflection spectrum in the UV range (Figure 3) nor protected the leaves from strong UV stress under either field or laboratory conditions (Figure 8). We speculate that several reasons might account for these observations. First, the different results were attributed to the different compounds, plants, or UV treatments. Second, in addition to anthocyanin, the changes in other metabolites (Figure 5) might also affect the absorption of UV light. Third, we used Fv/Fm, an indicator for the status of the photosynthetic apparatus, especially photosystem II [29], to reflect the UV damage to leaves in this study. However, the chloroplast might not be the most sensitive organelle in plants in response to UV stress. Thus, the UV stress damage might not be reflected by the Fv/Fm values over time. Further studies are needed to clarify this phenomenon.

In general, anthocyanin may protect plants from high light stress by absorbing excessive visible light despite reducing photosynthesis. However, anthocyanin cannot protect plants from high UV light stress.

## 4. Materials and Methods

### 4.1. Plant Materials and Treatments

A transcription factor, MdMYB10, which has been demonstrated to regulate the synthesis of anthocyanin in apple plants [2], was overexpressed in ‘Galaxy Gala’ (WT) to obtain the plants with anthocyanin accumulation. Transformation of ‘Galaxy Gala’ was performed as previously described [33,34]. Briefly, ‘Galaxy Gala’ shoots were subcultured on proliferation medium [35] at 4-week intervals, and were then transferred to leaf expansion medium (MS supplemented with 0.1 mg/L naphthyleneacetic acid and 8.3 mg/L 6-(γ,γ-dimethylallylamino) purine 4 weeks before transformation. The youngest unfolded leaf was exercised from in vitro shoots, and the leaf was wounded using a pair of non-traumatic forceps [36] and co-cultivated with EHA105 (pCH32) harboring the binary vector pSAK277-*MdMYB10* [2]. After co-cultivation, the leaves were washed with distilled water containing half-strength MS salt mixture and cefotaxim sodium (500 mg/L) to remove excess inoculum and placed on regeneration medium containing 100 ppm kanamycin. The regenerated shoots were propagated in vitro. Wood buds of two transgenic lines (L8 and L9) and wild type (WT) were grafted onto M9-T337 rootstock and grown in the field for 2 years. During the growth period, no blossoms were found on any of the trees. The trees were subjected to standard horticultural management with pest and disease control.

For the high light treatment with artificial light sources, fully expanded leaves were sampled in the morning, punched into leaf discs (diameter 1.5 cm). The abaxial surface of leaf discs was closely attached to four layers of wet gauze, and then treated under high light for 0, 1, 2, and 3 h at room temperature. The UV-B, UV-A, blue, green, and red lights were provided by custom-made LED-light sources, with a wavelength of approximately 280, 370, 450, 520, and 670 nm, respectively (Appendix A). A custom-made light source with mixed red, green, and blue LED-light beads was considered as white light. During the treatments, the light intensities used to irradiate leaf discs were approximately 90 μW cm^−2^ for UV-B, 3000 μW cm^−2^ for UV-A, and 2000 μmol m^−2^ s^−1^ for visible light. For each treatment at each sampling time, five replicates were used for the chlorophyll a fluorescence transient and 820 nm modulated reflection measurements with M-PEA (Hansatech, Norfolk, UK) as described by Gao et al. [29]. The wavelength of the red light source of M-PEA is 627 ± 10 nm, which is not absorbed by anthocyanin, thereby excluding any possible interference of anthocyanin in the measurements [37]. The maximum quantum yield of photosystem II (Fv/Fm) was calculated as Fv/Fm = (Fm − Fo)/Fm, where Fm and Fo are the maximum and minimum chlorophyll fluorescence intensity, respectively. Before performing the measurements, leaves were dark-adapted for 1 h. The leaf samples were also collected and stored at −80 °C for biochemical analysis. Fully expanded leaves and new young branches were also collected from the southern canopy of the plants in the field at 8:00 in the morning or 14:00 in the afternoon, transferred to the laboratory, and then immediately used for measurements of chlorophyll a fluorescence transients after dark adaption. During plant growth, the maximum solar UV-B, UV-A, and visible light intensity were 60 μW cm^−2^, 1100 μW cm^−2^, and 1800 μmol m^−2^ s^−1^, respectively, on sunny days.

### 4.2. Metabolite Content and Enzyme Activity Analysis

Anthocyanin, chlorophyll, and xanthophyll cycle pigments were extracted and assayed as described by Zhang et al. [38]. A 20A HPLC (Shimadzu, Kyoto, Japan) was used for the anthocyanin and xanthophyll cycle pigment content assay, and a UV-2450 spectrophotometer (Shimadzu) was used for the chlorophyll content determination. Isolated cyanidin-3-*O*-galactoside with a purity over 98% at different concentrations was used to generate a standard curve to calculate the concentration of anthocyanin in plants.

Non-structural carbohydrates, organic acids, and free amino acids were assayed as described by Li et al. [39], using an Agilent 7890A GC/5975C MS (Agilent Technology, Palo Alto, CA) and an Agilent 1100 HPLC equipped with a fluorescence detector (Agilent Technology), respectively. Sugar phosphates and other highly polar intermediates were determined on an AB Sciex QTRAP 4000 Mass Spectrometer (Applied Biosystems/MDS Sciex, Waltham, MA) linked to a Shimadzu HPLC according to Ma et al. [40].

Contents of ascorbate and glutathione as well as the activity of superoxide dismutase (SOD), ascorbate peroxidase (APX), monodehydroascorbate reductase (MDHAR), dehydroascorbate reductase (DHAR), and glutathione reductase (GR) were assayed as described by Zhang et al. [38]. The activity of ribulose-1,5-bisphosphate carboxylase/oxygenase (Rubisco), glyceraldehyde-3-phosphate dehydrogenase (GAPDH), phosphoribulokinase (PRK), sucrose phosphate synthase (SPS), and aldose-6-phosphate reductase (A6PR) were assayed according to previous method [41].

### 4.3. Photosynthesis, Leaf Reflection Spectra, and Stomatal Phenotype Assay

On sunny days in June, light-response and CO_2_-reponse curves of photosynthetic rates were measured from 9:00 to 11:00 a.m. using a LI-6800 portable photosynthesis system (Li-Cor Biosciences, Lincoln, NE, USA). For light-response curves measurement, the photosynthetic irradiation light (red:blue = 90:10) was provided by the light source of the LI-6800, changing the intensity from high to low in the following sequence: 1800, 1400, 1000, 750, 500, 250, 150, 100, and 50 μmol m^−2^ s^−1^. Respiration was measured without any irradiation of the leaves. Apparent quantum yield was calculated from the initial slope of the light-response curves. The CO_2_ concentration was set as 400 μmol mol^−1^. For CO_2_-response curves measurement, the CO_2_ concentrations were changed in the following sequence: 400, 300, 200, 100, 400, 550, 700, 900, 1200, and 1400 μmol mol^−1^. The light intensity was set at 1600 μmol m^−2^ s^−1^. Fully expanded leaves were kept in each intensity or CO_2_ concentration for at least 3 min until the photosynthetic rates became stable. Leaf reflection spectra were assayed using a UV-2600 spectrophotometer equipped with an IS-2600 plus integrated sphere (Shimadzu). The phenotypic assay of stomata was carried out as described by Li et al. [42] using an Olympus BX-63 microscope (Olympus Corp., Tokyo, Japan) with cellSens Entry software (Tokyo, Japan).

### 4.4. RNA-Seq Analysis

Total RNA was isolated from fully expanded leaves using the SDS-phenol method [43] and subjected to Novogene in Beijing (https://en.novogene.com (accessed on 3 December 2019)) for library construction and RNA sequencing with an Illumina NovaSeq 6000. The clean data for each sample were mapped to the GDDH13 genome [44] with Hisat2, and the corresponding read counts were analyzed by Deseq2 to select differentially expressed genes. The resulting *p* values were adjusted using Benjamini and Hochberg’s approach for controlling the false discovery rate, padj < 0.05 and the absolute value of log_2_(fold change) ≥ 1 were set as the threshold for significantly different expression. Kyoto Encyclopedia of Genes and Genomes (KEGG) pathways of differentially expressed genes were determined using the clusterProfiler R package at the website https://www.kegg.jp/kegg/pathway (accessed on 15 March 2020).

### 4.5. Statistical Analysis

All data are presented as means ± SE and were analyzed with the *t*-test or least significant difference (LSD) test using SPSS 16.0 software (SPSS, Chicago, IL, USA). Heatmap representations were produced using the TBtools [45].

## Figures and Tables

**Figure 1 ijms-23-12616-f001:**
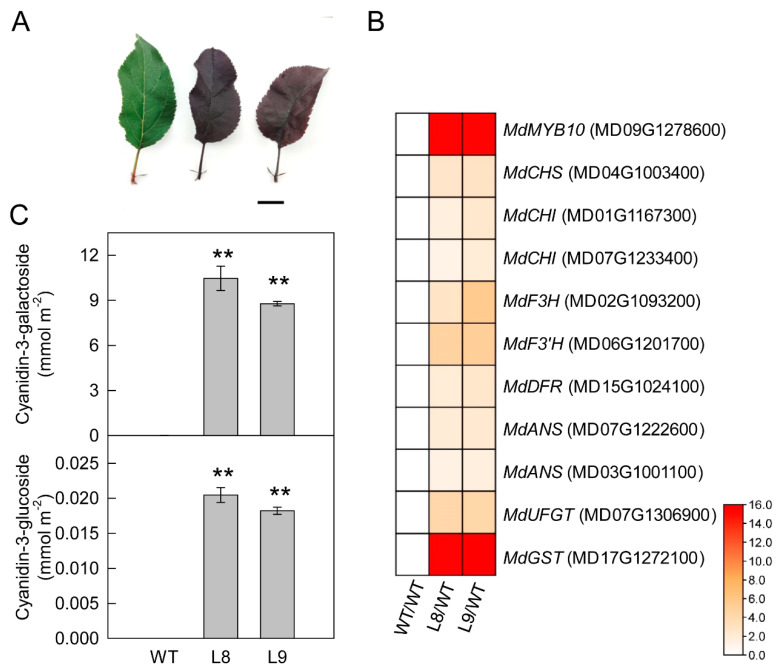
Leaf phenotypes (**A**), differentially expressed genes involved in anthocyanin synthesis (**B**), and contents of anthocyanins (**C**) in the leaves of ‘Galaxy Gala’ (WT) and its transgenic lines (L8, L9) with overexpression of *MdMYB10*. Panel A, scale bar = 2 cm; Panel B, data were extracted from the RNA-sequence assay between transgenic lines and wild type, the RNA-sequence data are shown in Appendix A, with three biological replicates; Panel C, five biological replicates were performed for each point. For the statistical analysis, “**” above the bars indicates a significant difference at *p* < 0.01, *t*-test. *CHS, chalcone synthase*; *CHI, chalcone isomerase*; *F3H, flavanone 3-hydroxylase*; *F3′H, flavonoid 3′-hydroxylase*; *DFR, dihydroflavonol 4-reductase*; *ANS, anthocyanin synthase*; *UFGT, UDP-glycose: flavonoid glycosyltransferase*; *GST, glutathione S-transferase*.

**Figure 2 ijms-23-12616-f002:**
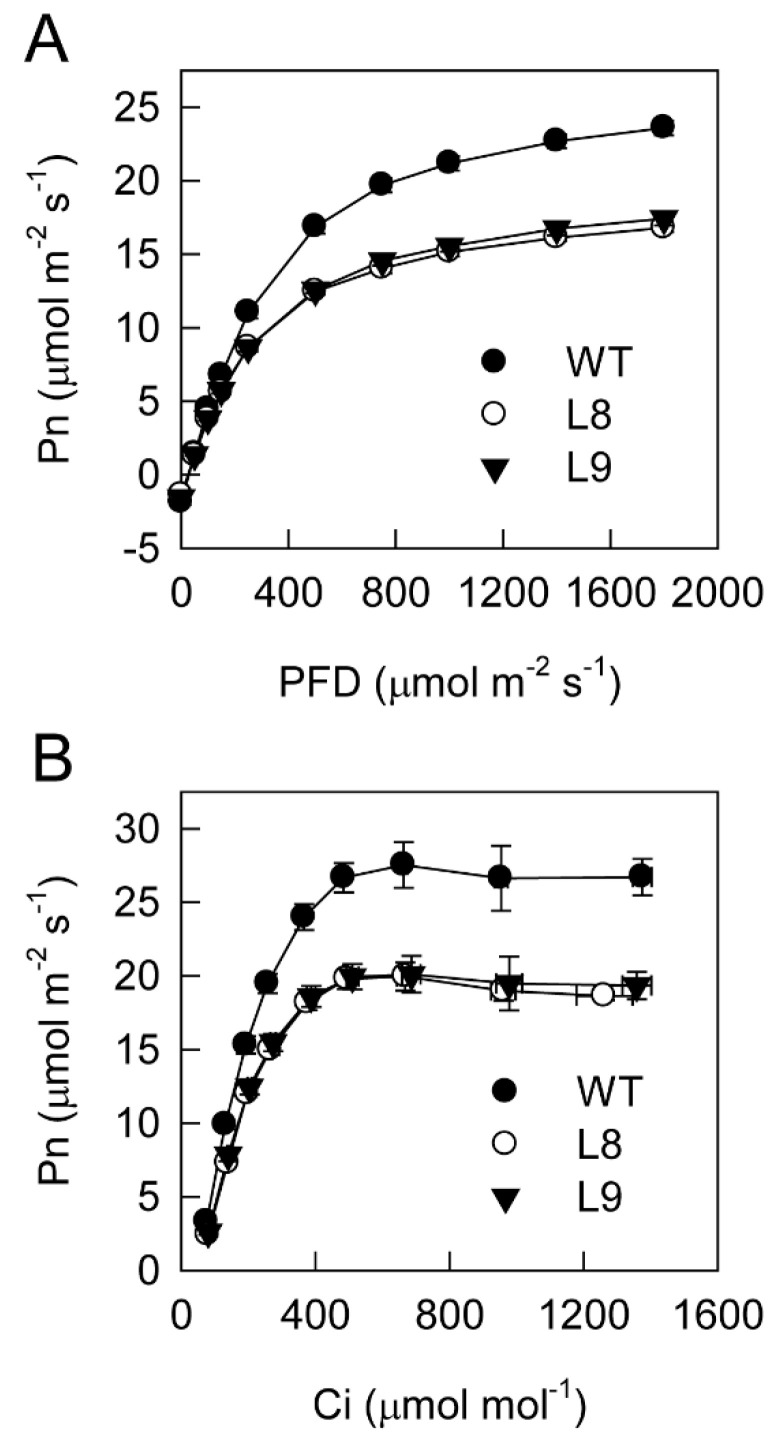
Photosynthesis light response (**A**) and CO_2_ response (**B**) curves for the leaves of ‘Galaxy Gala’ (WT) and its transgenic lines (L8, L9) with overexpression of *MdMYB10*. Five biological replicates were performed for each point. Pn, Net photosynthetic rate; PFD, photon flux density; Ci, intercellular CO_2_ concentration.

**Figure 3 ijms-23-12616-f003:**
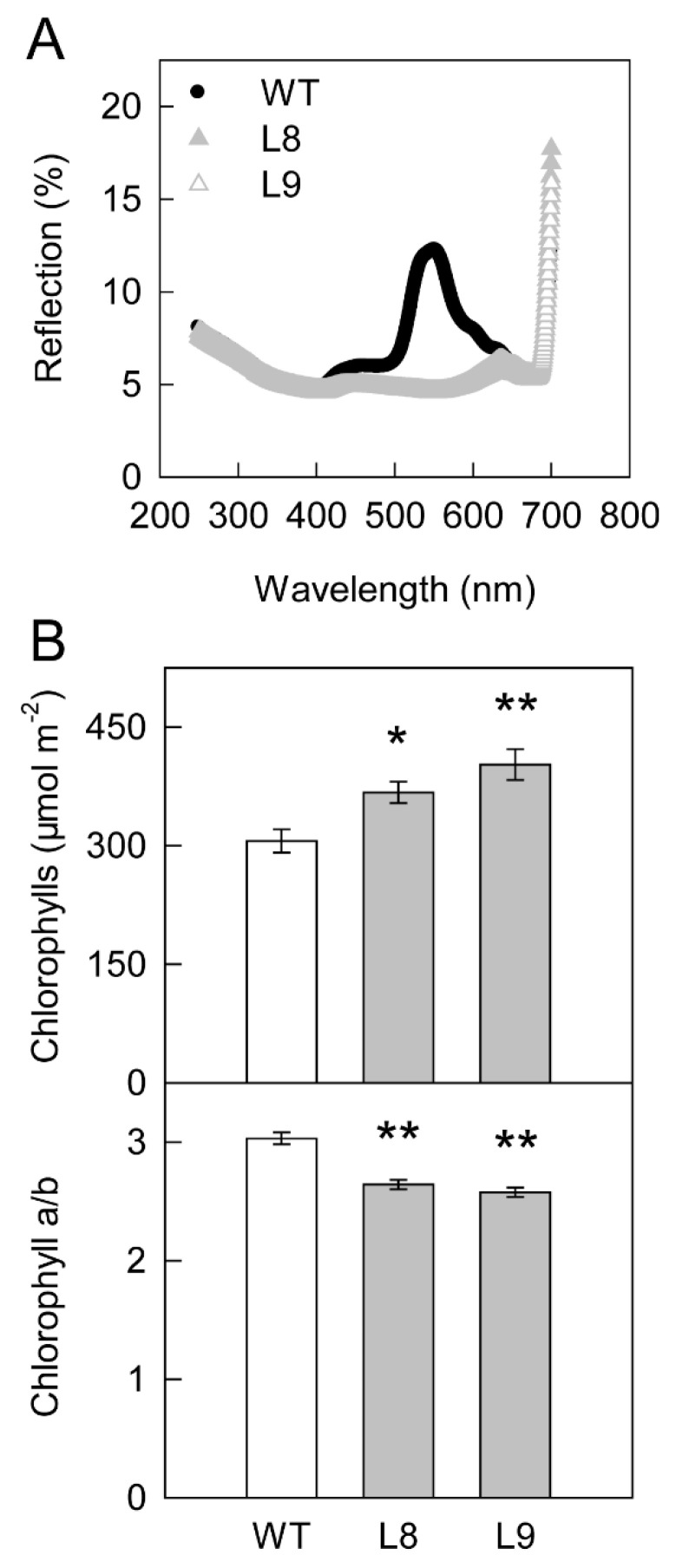
Leaf reflection spectra (**A**) and chlorophyll content, as well as the ratio of chlorophyll a to b (**B**) in leaves of ‘Galaxy Gala’ (WT) and its transgenic lines (L8, L9) with overexpression of *MdMYB10*. Five biological replicates were performed for each point. For statistical analysis, “*” or “**” indicates a significant difference at *p* < 0.05 or *p* < 0.01, respectively, *t*-test.

**Figure 4 ijms-23-12616-f004:**
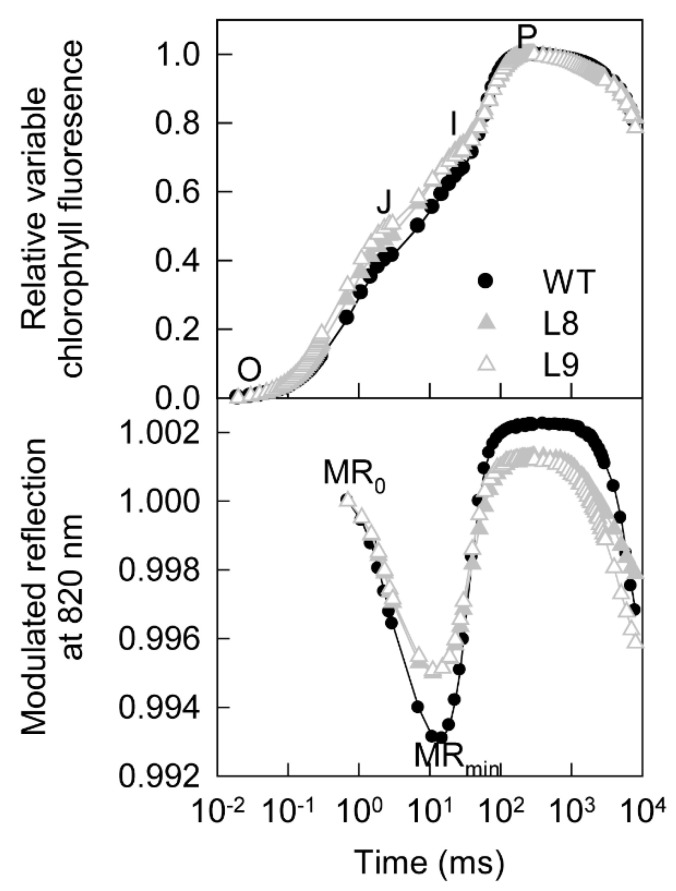
Relative variable chlorophyll fluorescence transient [(Ft − Fo)/(Fm − Fo)] and modulated 820 nm reflection (MRt/MRo) curves for the leaves of ‘Galaxy Gala’ (WT) and its transgenic lines (L8, L9) with overexpression of *MdMYB10*. Five biological replicates were performed for each point.

**Figure 5 ijms-23-12616-f005:**
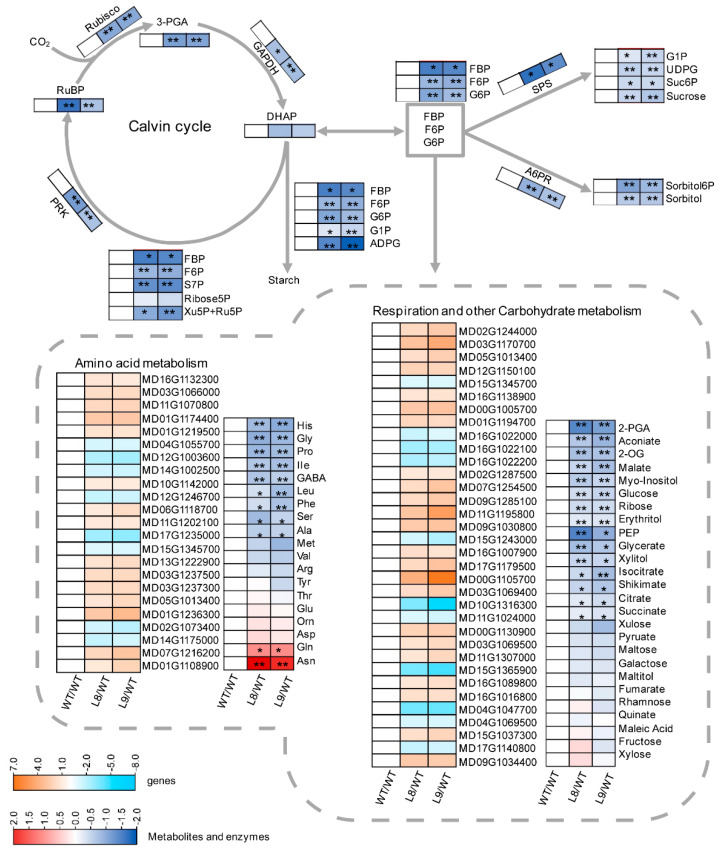
Comparison of expressions of key genes, activities of key enzymes, and contents of key metabolites involved in photosynthesis and other related metabolisms between the leaves of ‘Galaxy Gala’ (WT) and its transgenic lines (L8, L9) with overexpression of *MdMYB10*. As none of the gene involved in photosynthesis showed different expression levels between transgenic and WT leaves (Appendix A), only genes involved in respiration and other carbon and nitrogen metabolism were shown in Figure 5. For the expression of each gene, please see Appendix A. The absolute values of metabolites and enzyme activities are shown in Appendix A. Rubisco, ribulose-1,5-bisphosphate carboxylase/oxygenase; GAPDH, glyceraldehyde-3-phosphate dehydrogenase; PRK, phosphoribulokinase; A6PR, aldose-6-phosphate reductase; 3-PGA, 3-phosphoglycerate; DHAP, dihydroxyacetone phosphate; FBP, fructose-1,6-bisphosphate; F6P, fructose-6-phosphate; S7P, Sedoheptulose 7-phosphate; Ribose5P, ribose 5 phosphate; XU5P, xylulose-5-phosphate; RU5P, ribulose 5-phosphate; RUBP, ribulose-1,5-bisphosphate; G6P, glucose-6-phosphate; G1P, glucose-1-phosphate; ADPG, adenosine diphosphate glucose; UDPG, uridine diphosphate glucose; Suc6P, sucrose-6-phosphate; sorbitol-6-P, sorbitol-6-phosphate; 2-PGA, 2-phosphoglycerate; 2-OG, 2-oxoglutarate; PEP, phosphoenolpyruvate. WT/WT, L8/WT, and L9/WT denote the values of the log_2_ fold change between genotypes. For the transcriptome assay, three biological replicates were performed; for the enzyme activities and metabolomics assay, five biological replicates were conducted. For the statistical analysis, “*” or “**” indicates a significant difference at *p* < 0.05 or *p* < 0.01, respectively, *t*-test.

**Figure 6 ijms-23-12616-f006:**
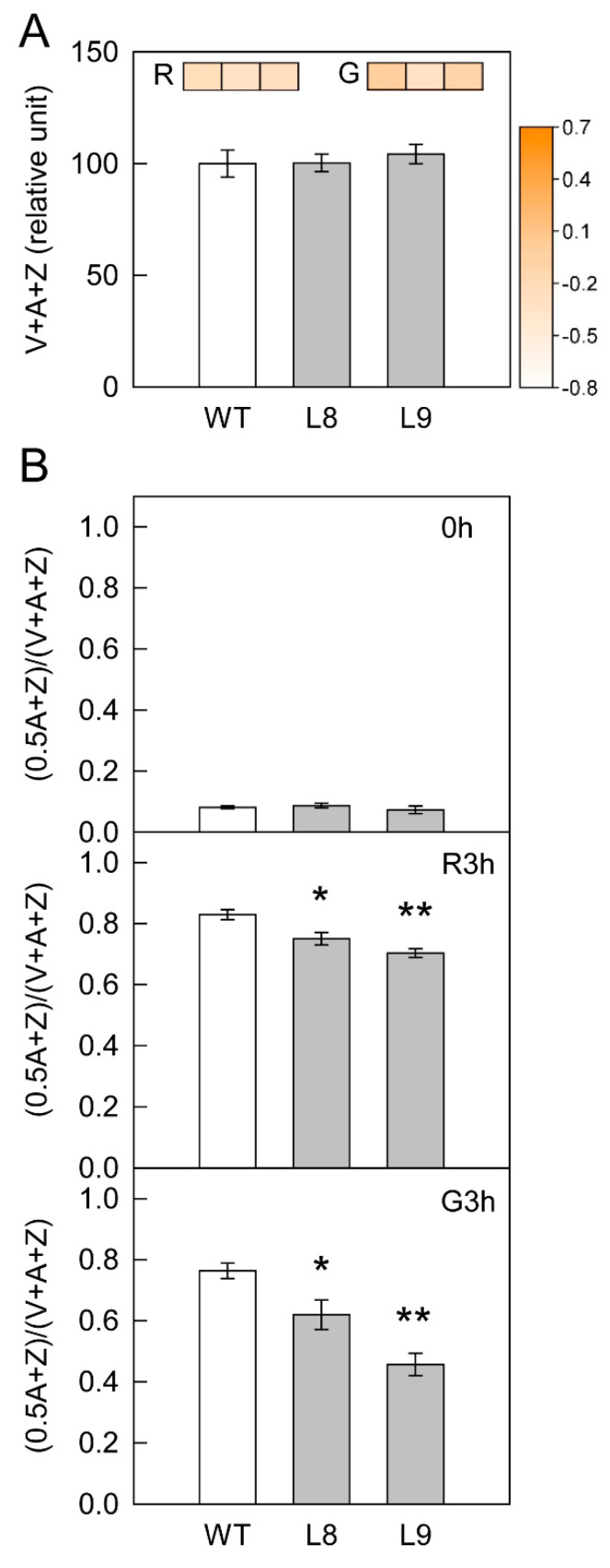
Xanthophyll cycle pool size (V + A + Z) (**A**) and its de-epoxidation [(0.5A + Z)/(V + A + Z)] (**B**) in leaves of ‘Galaxy Gala’ (WT) and its *MdMYB10* overexpression transgenic lines (L8, L9) before and after strong red or green light treatment. Panel A, the inserted heat maps with blocks from left to right are the values of the log_2_ fold change of the xanthophyll cycle pool size of WT, L8, and L9 before and after strong red (R) or green (G) light treatment, respectively. Five biological replicates were performed for each point; Panel B, ‘0 h’, ‘R3h’ and ‘G3h’ represented before (0 h) and after strong green (G3h) or red (R3h) light treatment for 3 h, respectively. For the statistical analysis, “*” or “**” indicates a significant difference at *p* < 0.05 or *p* < 0.01, respectively, *t*-test.

**Figure 7 ijms-23-12616-f007:**
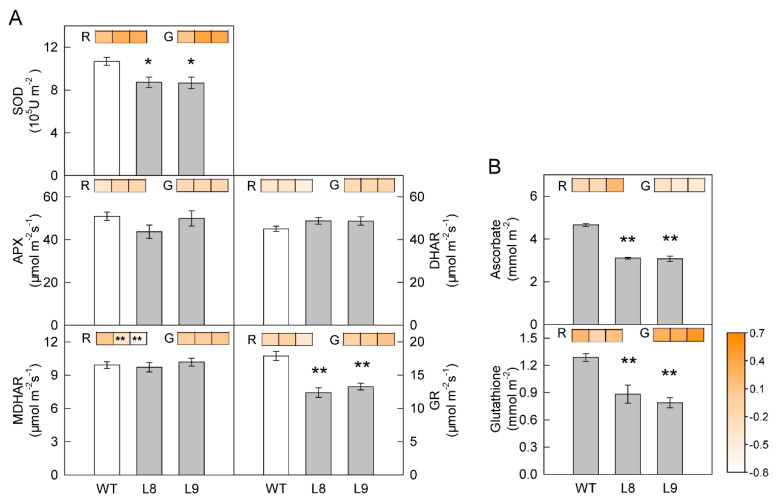
Antioxidant enzyme activity (**A**) and metabolite content (**B**) in leaves of ‘Galaxy Gala’ (WT) and its *MdMYB10* overexpression transgenic lines (L8, L9) before and after strong red or green light treatment. The inserted heat maps with blocks from left to right are the values of the log_2_ fold change in enzyme activity or metabolite content of WT, L8, and L9 before and after strong red (R) or green (G) light treatment, respectively. Five biological replicates were performed for each point. SOD, superoxide dismutase; APX, ascorbate peroxidase; MDHAR, monodehydroascorbate reductase; DHAR, dehydroascorbate reductase; GR, glutathione reductase. Five biological replicates were performed for each point. For the statistical analysis, “*” or “**” indicates a significant difference at *p* < 0.05 or *p* < 0.01, respectively, *t*-test.

**Figure 8 ijms-23-12616-f008:**
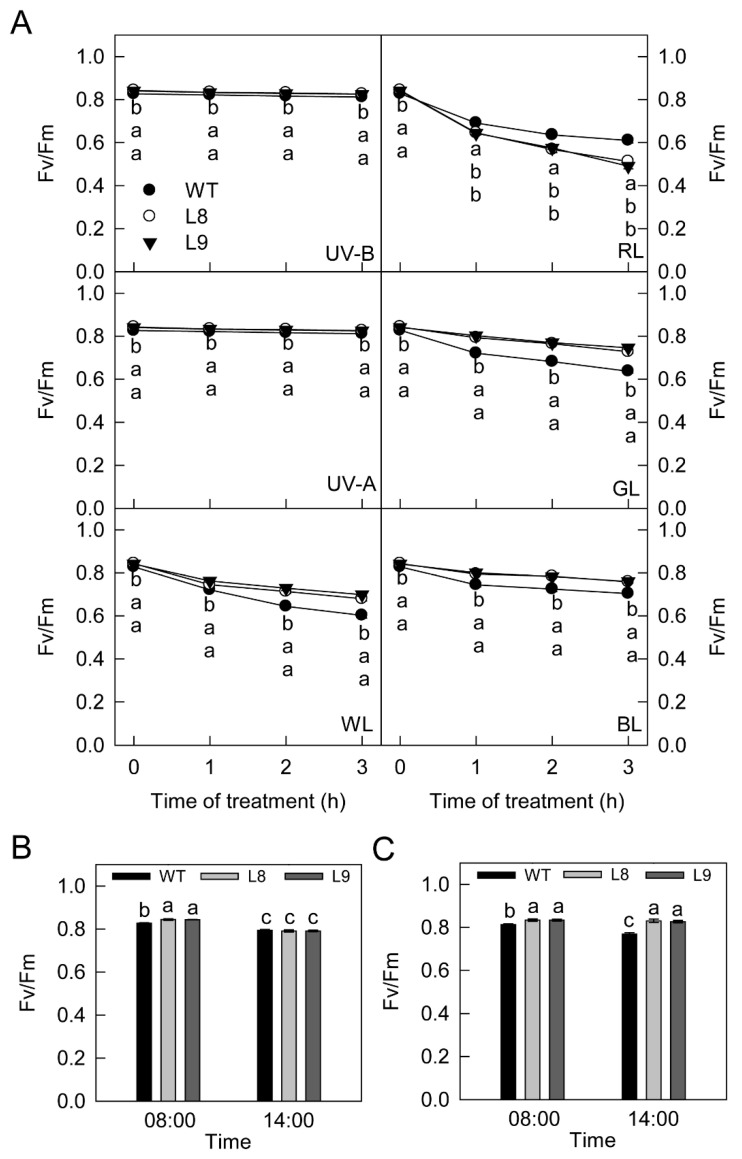
The maximum quantum yield of photosystem II (Fv/Fm) in leaves of ‘Galaxy Gala’ (WT) and its transgenic lines (L8, L9) with *MdMYB10* overexpression after treatment with different artificial light sources under laboratory conditions (**A**) or exposure to sunlight in the field (**B**), and in young branch barks of wild type and transgenic lines exposed to sunlight in the field (**C**). WL, white light, RL, red light; GL, green light; BL, blue light. The relative changes in Fv/Fm in Panel A are shown in Appendix A. Five biological replicates were performed for each point, with two leaves or branches from one plant assayed for each replicate. For the statistical analysis, different lowercase letters below symbols or above bars indicate a significant difference at *p* < 0.01, LSD.

## Data Availability

All data that support the findings of this study are available from the corresponding author upon reasonable request.

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
