# Peer review of "Anthocyanin Accumulation Provides Protection against High Light Stress While Reducing Photosynthesis in Apple Leaves"

_ijms, 2022, doi:10.3390/ijms232012616_

Round 1

Reviewer 1 Report

The manuscript of Zhao et al describes the role of highly accumulated anthocyanin on the protection of apple leaves from high light irradiation. This work applied multi-methods to explore the impact of high anthocyanin on photosynthesis, gene expression, metabolisms, antioxidant enzyme activity. The conclusion is interesting that the high anthocyanin will reduce photosynthesis. However, it looks like all the figures, especially the words in the figures are bad quality, or the charts (Figure 4) are fuzzy. Overall, this study is well designed, and the experiments are well conducted. I support this manuscript for publication in IJMS.

Minor comments:

1.      91-92, all the gene names shall provide the full name.

2.      93, what means and its subsequent transfer from the cytosol to vacuoles in transgenic leaves (Fig. 1)

3.      112, Please give some reason/background why gave the description Dark respiration was comparable between WT and transgenic red leaves (Fig. 2A).

4.      123, Pn is the abbreviation of what?

Reviewer 2 Report

International Journal of Molecular Sciences reviewer comments

This is a very interesting paper.

A few points need attention.

Line 113. Was the Photon Flux Density measured over the 400-700 nm wavelength range? The wavelength range in figure 3A starts at ~ 250 nm.

Lines 116-117. A little more detail on how the apparent quantum yield was measured would be useful. I do not have access to reference 29 to which the reader is referred in Materials and Methods.

Figure 2A. Are the photosynthetic rates for L8 and L9 significantly different from controls at PFDs below 100 μmol m-2 s-1?

Figure 3A gives values for reflectance. Was transmittance measured? This would be important leaves lower in the canopy if the leaf area index is greater than 1.

Line 206. α-ketoglutarate is more commonly represented as 2-oxoglutarate.

Lines 227-231. Interesting method.

Lines 274-275. Does this much lower rate of photosynthesis in non-leaf tissue  relate to apple, or is it more widely applicable? See Henry et al. 2002 MDPI Biology 9: 438.

Line 287. ‘substantial’ rather than ‘largely’.

Lines 311-321. Good point.

Lines 348-349, 362-363. I appreciate that UV is commonly reported in energy units, while photosynthetically active radiation in commonly reported in photon units. However, it would be helpful to the reader to express UVA and UVB in photon units.

Reviewer 3 Report

Reviewer comments

The current research provides more data to improve our understanding of the role of anthocyanin on photoprotection under high light stress. The manuscript is well written, easy to follow and understand. However, there are a few major and minor issues that need to be addressed before the manuscript can be published.

Major revision:

Although the findings are well discussed, there’s no clear conclusion. The authors need to state clearly the conclusion(s) or major takeaway(s) from their research to enable readers determine the significant contribution to knowledge averred by the current study.

Minor revisions:

Fig S2: check the spelling of phlorizin (‘h’ is missing)

Line 136: check the spelling of 'reflection' (‘l’ is missing)

Figure 6: The following notations, ‘0h’, ‘G3h’ and ‘R3h’, used in panel B should be defined in the figure description/caption.

L248-249: Figure S4 should be Figure S5.
